# Competitive analysis of online revenue management with two hierarchical resources and multiple fare classes

**Guanqun Ni** [ORCID] *

Business School, Shandong University of Technology, Zibo, Shandong, China

* guanqunni@163.com

## Abstract

Resource allocation problem is one of key issues in the field of revenue management. The traditional models usually rely on some restrictive assumptions about demand information or arrival process, which is sometimes out of line with reality. To overcome this shortcoming, the method of competitive analysis of online algorithms, which eliminates the need for the assumptions on demand and arrivals, is adopted to deal with the quantity-based revenue management problem. The current model in this paper considers two downgrade compatible levels of resources. Given the capacities and fares of both levels of resources, the objective is to accept appropriate customers and assign them to appropriate resources so as to maximize revenues. Compared with the existing literature, this paper generalizes the concerned resource allocation problem by considering multiple fares for each level of resources. From the perspective of online algorithms and competitive analysis, both an upper bound and an optimal online strategy are derived in this paper.

## Introduction

In the field of revenue management (RM), one of the key issues is to deal with the *quantity-based single-leg* problem, where the decision maker allocates inventory of a kind of resource to a demand stream and each customer in the stream has a requested fare drawn from a given set of fare classes [1]. For example, given a flight with 100 available seats and three fare classes, e.g. $1000, $800, and $500, the airline has to decide how many seats to be allocated to different fare classes in order to maximize revenues. If the demand of each fare class is known in advance, the resource allocation problem is easy to solve, i.e., allocating seats to higher fare class as far as possible. Since the demand information is usually uncertain in practice, the majority of successful RM implementations rely heavily on accurate demand forecasting [2, 3]. Meanwhile, the traditional theoretical models often rely on some restrictive and unrealistic assumptions about demand information or arrival process such as independence and stationarity [4].

However, in the case of short life products or situations where demand history is not necessarily available, to obtain accurate characterization of demand is a challenge [5, 6]. Even in the fields of both airlines and hotels, where the application of RM techniques is the most

**Citation:** Ni G (2022) Competitive analysis of online revenue management with two hierarchical resources and multiple fare classes. PLoS ONE 17(10): e0276530. https://doi.org/10.1371/journal.pone.0276530

**Data Availability Statement:** All relevant data are within the paper.

**Funding:** This work is partially supported by the MOE Layout Foundation of Humanities and Social Sciences under grant 20YJA630049 and Startup

Program of Doctor Scientific Research in Shandong University of Technology under grant 722001. The funders had no role in study design, data collection and analysis, decision to publish, or preparation of the manuscript.

**Competing interests:** The author has declared that no competing interests exist.

successful, obtaining high quality forecasts from historical data is not easy. To relax the assumptions for both demand information and arrival process, we study the quantity-based RM problem from the perspective of *online algorithms and competitive analysis* [7]. This method eliminates the need for either of these assumptions, which is particularly suitable for decision makers who have to respond to events over time, like passengers booking tickets one at a time in the airline scenario.

Besides, the single-leg RM models in previous studies usually assume that the resource is non-hierarchical, which is sometimes out of line with reality. The quantity-based RM problem dealing with hierarchical resources is very common in practice. To see the importance of this scenario, consider an airline that provides potential passengers with two classes of tickets (i.e. hierarchical resources), economy class and business class, for one flight. The full fare of economy class ticket is undoubtedly lower than that of business class ticket. In order to increase revenue, airlines usually sell both economy class and business class at different discounts based on market segmentation. Generally, the lowest discount fare of business class is probably still higher than the full fare of economy class. Suppose that the set of all fares, including full fares and discount fares, is $\{f_1, f_2, \ldots, f_m\}$ and $f_1 < f_2 < \ldots < f_m$. Without loss of generality, let $f_k$ ($1 \leq k < m$) be the full fare of economy class and $f_m$ be the full fare of business class. Obviously, $f_1$ and $f_{k+1}$ are the lowest discount fares for economy class and business class, respectively. Note that, all fares are predetermined and invariable. Facing a request of customer with reservation price $p$ for the ticket, if $f_i \leq p < f_{i+1}$ for some $i \leq k$, we call the customer economy class passenger and the airline can charge at most $f_i$ no matter which class, economy or business, is assigned to this customer. While, if $f_i \leq p < f_{i+1}$ for some $i \geq k + 1$ (specially, $f_{m+1} = +\infty$ when $i = m$), we call the customer business class passenger and the airline has two choices, assigning the customer to business class charging $f_i$ or assigning the customer to economy class charging $f_k$. Given the capacities of both economy class and business class, the airline has to make decisions on assigning how many customers with different reservation price to each class. Since the airline cannot know demand information in advance, the decision-making problem is indeed of an online fashion. Other notable applications are hotel booking with heterogenous rooms and car rental with different classes of autos. The major issue in these scenarios is how to accept appropriate customers and assign them to appropriate resources so as to maximize revenues, i.e., the payments of all accepted customers.

Our contributions can be summarized as follows. We propose a new online strategy for the quantity-based RM problem with two levels of resources where the resources are downgrade compatible. The current model in this paper generalizes the concerned problem by considering multiple fares for each level of resources. From a practical point of view, it is difficult, if not impossible, to exactly know future demand; that is the decision maker has to make resource allocation decisions in a learning-by-doing manner based on the previous decisions and outcomes. For this learning-by-doing decision-making problem, we focus on online strategy and use standard competitive analysis to evaluate an online strategy. In this paper, an upper bound of competitive ratio for the problem is derived. We also propose an optimal online strategy whose competitive ratio matches the upper bound.

The quantity-based resource allocation problem has long been studied in the RM literature. Littlewood was the first to formulate the single-leg RM problem with two fare classes [8]. He assumed that resource is sold in a low-before-high (LBH) manner; that is, demand in lower fare class arrives earlier. Given the probability distribution of demand for the high fare class, Littlewood determined the booking limit for the low fare class. With the same assumptions, Belobaba extended Littlewood's model to multiple fare classes [9, 10]. The LBH assumption is widely applied in the traditional single-leg RM studies [11–13]. There are several other studies on the single-leg RM problem with various assumptions on demand or arrivals, e.g. arriving at

non-overlapping intervals [14], a stochastic process [15], discrete time Markov Decision Process [16], or demand information on the choice behavior [17]. In practice, it is probably difficult to meet these assumptions on demand. Therefore, we need more robust approaches which do not rely heavily on demand information to deal with the quantity-based RM problem.

In the new stream of research, the works of [18–20] are the most related to our study, all of which adopt the approach of competitive analysis of online algorithms. Ball and Queyranne, to the best of our knowledge, were the first to deal with the single-leg resource allocation RM problem from the perspective of online algorithms and competitive analysis [18]. They defined the lower bounds on the best-possible algorithmic performance and designed optimal nested protection-level policies whose competitive ratios match the lower bounds. In their model, there is no any assumption on demand or arrivals. Also from the perspective of competitive analysis of online algorithms, Lan et al. studied the classical multiple fare single-leg RM problem with limited demand information where the decision maker only knows the lower and upper bounds of demand for each fare class [19]. That is, the work of [19] is more general than that of [18] by assuming the availability of limited demand information. The competitive analysis of online algorithms has also been applied for other single-leg RM problem under various situations in the literature [21–23]. All of these studies assume that the resource is non-hierarchical, which is sometimes out of line with reality. Like the example in the field of airlines, the quantity-based RM problem dealing with compatible hierarchical resources is very common. To overcome this shortcoming, in this paper we consider the single-leg RM problem with two levels of resources, which are downgrade compatible; that is the resource of high level can be used as low level, but the resource of low level cannot be used as high level. Recently, Ni et al. proposed an optimal online strategy for the simplest problem with two levels of resources, in which there is only one fare class for each level of resources [20]. However, taking the field of airlines as an example, there usually are multiple fare classes even for the same level of resources. In order to be more realistic, we further extend the work of [20] by considering more than one fare class for the same level of resources. It is worth noting that our model will degenerate to the *Multiple-Fare Classes* problem studied in [18] when there is only one level of resources available. Besides, the current model will degenerate to the *2-level-2-class* problem studied in [20] if there is only one fare class for each level of resources.

The rest of this paper is organized as follows. In the following section, we introduce the problem and describe our approach. Next, an upper bound and an optimal online strategy are presented from the perspective of online algorithms and competitive analysis. Finally, applications and future directions are discussed.

## Problem definitions

Consider the problem of a firm having two levels of resources, denoted by $R_1$ and $R_2$, to provide a kind of service for heterogeneous customers. Suppose that resource $R_i$ has a capacity of $L_i$ ($i = 1, 2$) where $L_i$ is an integer, and each customer only needs one unit capacity of resource. Although both levels of resources can satisfy customers' basic need, we assume that $R_2$ serves customers better or more comfortably than $R_1$; that is, the level of $R_2$ is higher than $R_1$. As described in the airline scenario, the firm indeed sells both levels of resources at different discounts based on market segmentation and predetermines the fare of each class. Let $\{f_1, f_2, \ldots, f_m\}$ be the set of all fares and $f_1 < f_2 < \cdots < f_m$. Without loss of generality, let $f_k$ ($1 \leq k < m$) be the full fare of unit $R_1$ and $f_m$ be the full fare of unit $R_2$. Facing a request of customer with reservation price $p$ for unit resource, if $f_i \leq p < f_{i+1}$ for some $i \leq k$, the firm can charge at most $f_i$ no matter which level of resources, $R_1$ or $R_2$, is assigned to this customer. While, if $f_i \leq p < f_{i+1}$ for some $i \geq k + 1$ (defining $f_{m+1} = +\infty$ for completeness), the firm has two choices, assigning the

**Table 1. The symbols of the resource allocation model.**

| Symbol | Description |
|---|---|
| $R_i$ | the resource of level $i$ for $i$ = 1, 2. |
| $L_i$ | the capacity of resource $R_i$. |
| $f_k$ | the full fare of unit resource $R_1$. |
| $f_m$ | the full fare of unit resource $R_2$. |
| $f_i$ | the discount fare of resource $R_1$ ($R_2$) if $1 \leq i < k$ ($k < i < m$). |
| $p$ | the reservation price of customer for the resource. |
| $C_i$ | the class of customers with reservation price belonging to $[f_i, f_{i+1})$. |
| $t = L_1/L_2$ | the ratio of capacities of two levels of resources. |
| $r_i = f_i/f_{i+1}$ | the ratio of adjacent fares for $1 \leq i < m$, specially, $r_0 = 0$ and $r_m = 1$. |

customer to $R_2$ charging $f_i$ or assigning the customer to $R_1$ charging $f_k$. According to their reservation prices, customers are also divided into $m$ classes, denoted by class $C_i$ ($i$ = 1, 2, ..., $m$), and the reservation price of customers of class $C_i$ is within the range of $[f_i, f_{i+1})$. Given the capacity of each level of resources, the objective of the firm is to choose the most profitable customers and assign them to the right resources to maximize the resulting revenue. All of the main symbols that are used in the resource allocation model are listed in Table 1.

If the demand information is known in advance, then it is an offline problem and there is an intuitional revenue-maximization strategy, i.e., assigning customers with higher reservation price to higher level of resources as far as possible. Namely, an optimal offline strategy first assigns customers of class $C_m$ to resource $R_2$ and then to resource $R_1$ until either all customers of this kind are assigned or all resources are used. If there are some resources available after accepting all customers of class $C_m$, then accept and assign customers of class $C_{m-1}$ first to $R_2$ as far as possible and then to $R_1$, and so forth. This continues until all customers are assigned or all resources are used.

As mentioned in the introduction, we focus on an online version of the RM problem, wherein customers ask for the service one at a time and the decision on whether to accept the request as well as assigning it to which level of resources has to be made irrecoverably on its arrival without the information of further customers. For a revenue-maximization problem, the theoretical performance of an online strategy is measured by the ratio between the objective revenue achieved by the online strategy and the revenue achieved by an optimal offline strategy, which is called *competitive ratio* in the literature. Let $\Omega_A$ be the set of all possible input sequences to an online strategy $A$. For $\forall I \in \Omega_A$, let $V_A(I)$ be the objective revenue achieved by $A$ facing input $I$ and let $V_{OPT}(I)$ be the objective revenue achieved by an optimal offline strategy. The competitive ratio of $A$ is defined as $c_A = \inf_{I \in \Omega_A} \frac{V_A(I)}{V_{OPT}(I)}$. An online strategy $A^*$ is the best one if the competitive ratio of $A^*$ is $c_{A^*} = \sup c_A$, where $\sup c_A$ is called the *upper bound* of the competitive ratio.

## Competitive analysis for the online RM problem

This section gives detailed competitive analysis for the online RM problem with two levels of resources and multiple fare classes. First, an online strategy is designed in the following, and then an upper bound for the proposed online RM problem is derived.

### An online nested protection strategy

For notational convenience, let $t = L_1/L_2$ be the ratio of the capacities and $r_i = f_i/f_{i+1}$ be the ratio of the fares for $1 \leq i < m$ in the following. Virtually set $r_0 = 0$ and $r_m = 1$. We define

$\alpha = \dfrac{1}{\sum\limits_{i=1}^{k}(1-r_{i-1})+\sum\limits_{i=k+1}^{m}\frac{1-r_{i-1}}{1+t}}$. Because $0 \leq r_i < 1$ and $t > 0$, the denominator is larger than 1 and obviously $0 < \alpha < 1$.

For customers of class $C_j$, set $\Theta_j = \alpha(L_1 + L_2)\sum\limits_{i=1}^{j}(1 - r_{i-1})$ if $1 \leq j \leq k$; otherwise if $k + 1 \leq j \leq m$, set $\Theta_j = (1 - \alpha\sum\limits_{i=j}^{m}(1 - r_i))L_2$. Define $\Theta_0 = 0$ virtually. The following lemma can be derived from the definition of $\Theta_j$.

**Lemma 1** *Given the definition of $\alpha$ and the values of $\Theta_j$ as defined,*

*(1)* $\sum\limits_{i=1}^{j}f_i(\Theta_i - \Theta_{i-1}) = \alpha f_j(L_1 + L_2)$ *for* $1 \leq j \leq k$, *and*

*(2)* $f_{k+1}L_1 + \sum\limits_{i=1}^{j}f_i(\Theta_i - \Theta_{i-1}) = \alpha(f_kL_1 + f_jL_2)$ *for* $k + 1 \leq j \leq m$.

**Proof of Lemma 1.** For part (1) with $1 \leq i \leq j \leq k$, $\Theta_i - \Theta_{i-1} = \alpha(L_1 + L_2)(1 - r_{i-1})$ directly from the definition of $\Theta_j$, and thus

$$\sum_{i=1}^{j}f_i(\Theta_i - \Theta_{i-1}) = \alpha(L_1 + L_2)\sum_{i=1}^{j}f_i(1 - r_{i-1})$$

$$= \alpha(L_1 + L_2)\sum_{i=1}^{j}(f_i - f_{i-1})$$

$$= \alpha(L_1 + L_2)f_j.$$

That means part (1) is straightforward and independent of the specific value of $\alpha$. The reasoning of part (2), however, is non-trival. Firstly, considering the part of $\sum\limits_{i=1}^{j}f_i(\Theta_i - \Theta_{i-1})$ with $k + 2 \leq j \leq m$,

$$\sum_{i=1}^{j}f_i(\Theta_i - \Theta_{i-1}) = \sum_{i=1}^{k}f_i(\Theta_i - \Theta_{i-1}) + f_{k+1}(\Theta_{k+1} - \Theta_k) + \sum_{i=k+2}^{j}f_i(\Theta_i - \Theta_{i-1}).$$

From part (1), we have that $\sum\limits_{i=1}^{k}f_i(\Theta_i - \Theta_{i-1}) = \alpha f_k(L_1 + L_2)$. By the definitions of $\Theta_{k+1}$ and $\Theta_k$, because of $r_m = 1$,

$$\Theta_{k+1} - \Theta_k = (1 - \alpha\sum_{i=k+1}^{m}(1 - r_i))L_2 - \alpha(L_1 + L_2)\sum_{i=1}^{k}(1 - r_{i-1})$$

$$= L_2 - \alpha L_2\sum_{i=k+2}^{m}(1 - r_{i-1}) - \alpha(L_1 + L_2)\sum_{i=1}^{k}(1 - r_{i-1}).$$

By the definition of $\Theta_j$ for $k + 2 \leq i \leq j \leq m$,

$$\Theta_i - \Theta_{i-1} = (1 - \alpha\sum_{l=i}^{m}(1 - r_l))L_2 - (1 - \alpha\sum_{l=i-1}^{m}(1 - r_l))L_2 = \alpha L_2(1 - r_{i-1}).$$

And thus,

$$\sum_{i=k+2}^{j} f_i(\Theta_i - \Theta_{i-1}) = \alpha L_2 \sum_{i=k+2}^{j} f_i(1 - r_{i-1}) = \alpha L_2(f_j - f_{k+1}).$$

Now, we can derive the left side of the equation in part (2) as follows.

$$f_{k+1}L_1 + \sum_{i=1}^{j} f_i(\Theta_i - \Theta_{i-1})$$

$$= f_{k+1}L_1 + \alpha f_k(L_1 + L_2) + \alpha L_2(f_j - f_{k+1})$$

$$+ f_{k+1}(L_2 - \alpha L_2 \sum_{i=k+2}^{m}(1 - r_{i-1}) - \alpha(L_1 + L_2)\sum_{i=1}^{k}(1 - r_{i-1}))$$

$$= \alpha(f_k L_1 + f_j L_2) + f_{k+1}(L_1 + L_2)$$

$$+ \alpha f_{k+1}(r_k L_2 - L_2 - L_2 \sum_{i=k+2}^{m}(1 - r_{i-1}) - (L_1 + L_2)\sum_{i=1}^{k}(1 - r_{i-1}))$$

$$= \alpha(f_k L_1 + f_j L_2) + f_{k+1}(L_1 + L_2)$$

$$- \alpha f_{k+1}(L_1 + L_2)(\frac{L_2}{L_1 + L_2}\sum_{i=k+1}^{m}(1 - r_{i-1}) + \sum_{i=1}^{k}(1 - r_{i-1}))$$

$$= \alpha(f_k L_1 + f_j L_2) + f_{k+1}(L_1 + L_2) - \alpha f_{k+1}(L_1 + L_2)\frac{1}{\alpha}$$

$$= \alpha(f_k L_1 + f_j L_2).$$

Note that $\sum_{i=1}^{j} f_i(\Theta_i - \Theta_{i-1}) = \sum_{i=1}^{k} f_i(\Theta_i - \Theta_{i-1}) + f_{k+1}(\Theta_{k+1} - \Theta_k)$ when $j = k + 1$, and we can similarly derive the result of part (2) for $j = k + 1$. Thus, the proof of part (2) is completed, which gives us the lemma.

Upon an arrival of any customer, denote by $l_i$, $l_{i1}$, and $l_{i2}$ the number of customers of class $C_i$ that have been accepted by the online strategy and been assigned to resource $R_1$ and resource $R_2$, respectively. Let $l_i = l_{i1} = l_{i2} = 0$ for $1 \leq i \leq m$ initially. Now, we present an online nested protection strategy, denoted by *ONPS*, for the problem as follows.

**Online nested protection strategy ONPS.** *Upon an arrival of a customer of class $C_j$ with $1 \leq j \leq k$, accept the customer's service request and assign the accepted customer to resource $R_1$ if $\sum_{i=1}^{h} l_i < \Theta_h$ for any $j \leq h \leq k$ and $\sum_{i=1}^{m} l_{i1} < L_1$, or assign the accepted customer to resource $R_2$ if $\sum_{i=1}^{h} l_i < \Theta_h$ for any $j \leq h \leq k$, $\sum_{i=1}^{m} l_{i1} = L_1$, and $\sum_{i=1}^{h} l_{i2} < \Theta_h$ for any $k + 1 \leq h \leq m$; otherwise reject it.*

*Upon an arrival of a customer of class $C_j$ with $k < j \leq m$, accept the customer's service request and assign the accepted customer to resource $R_2$ if $\sum_{i=1}^{h} l_{i2} < \Theta_h$ for any $j \leq h \leq m$, or assign the*

*accepted customer to resource $R_1$ if $\sum_{i=1}^{h} l_{i2} = \Theta_h$ for some $h \in [j, m]$ and $\sum_{i=1}^{m} l_{i1} < L_1$; otherwise reject it.*

By the above description of strategy *ONPS*, it guarantees a reservation capacity for each class of customers, and all the reservation capacities are nested. More precisely, for the highest $j$ ($\leq k$) classes of customers whose reservation prices are no more than $f_k$, i.e. $C_{k-j+1}$, $C_{k-j+2}$, ..., $C_k$, strategy *ONPS* defines a protection reservation capacity equal to $\Theta_k - \Theta_{k-j}$. With a little difference, for the highest $j$ ($< m - k$) classes of customers whose reservation prices are more than $f_k$, i.e. $C_{m-j+1}$, $C_{m-j+2}$, ..., $C_m$, strategy *ONPS* defines a protection reservation capacity equal to $L_2 - \Theta_{m-j}$. In special, for the highest $m - k$ classes of customers, i.e. $C_{k+1}$, $C_{k+2}$, ..., $C_m$, strategy *ONPS* defines a protection reservation capacity just equal to $L_2 - (\Theta_k - L_1) = L_1 + L_2 - \Theta_k$. Notice that, applying *ONPS*, only the higher class of customers can occupy the protection capacity reserved for the lower class. The lower class of customers cannot occupy the protection capacity reserved for the higher class.

**Theorem 1** *For the online quantity-based RM problem with two levels of resources and multiple fares, strategy ONPS has a competitive ratio of $\alpha$.*

**Proof of Theorem 1**. Given any customer input instance $I$, let $l'_i$ be the total number of customers of class $C_j$ for all $j \leq i$ that are accepted by *ONPS*, and virtually set $l'_0 = 0$. Define $u = \max\{i | l'_i = \theta_i, 0 \leq i \leq m\}$ where $\theta_i = \Theta_i$ for $0 \leq i \leq k$ and $\theta_i = \Theta_i + L_1$ for $k + 1 \leq i \leq m$. Because *ONPS* considers eligible resource from $R_1$ to $R_2$ (or, from $R_2$ to $R_1$) when assigning the accepted customers of class $C_j$ for all $j \leq k$ (or, for all $j > k$), the definition of $u$ implies that all the customers of class $C_j$ in instance $I$ with $u < j \leq m$ are accepted by *ONPS* and assigned to the right resources resulting in as much revenue as possible. Let $V^{on}$ and $V^{opt}$ be the total revenue obtained by *ONPS* and by an optimal offline strategy *OPT* facing instance $I$, respectively. According to the status of $u$, there are four cases as discussed below.

**Case 1**. $u = 0$. This condition implies that all the customers in $I$ are accepted by *ONPS* and assigned to the right resources from the above analysis, and thus $V^{on} = V^{opt}$.

**Case 2**. $1 \leq u \leq k$. For this case, let $\Pi$ be the total revenue of the accepted customers each of which is of a reservation price strictly larger than $f_u$, and $\phi$ be the total number of these customers accepted by *ONPS*. We already know that strategy *ONPS* reserves a capacity of $\Theta_k - \Theta_u$ for all customers of classes $C_{u+1}$, ..., $C_k$ where $0 \leq u \leq k - 1$. Hence, $V^{on} \geq \sum_{i=1}^{u} f_i(\Theta_i - \Theta_{i-1}) + \Pi \geq \alpha f_u(L_1 + L_2) + \Pi$ where the second inequality is due to Lemma 1. *OPT* at best accepts all the $\phi$ customers with revenue larger than $f_u$ as *ONPS* does, and accepts the rest $L_1 + L_2 - \phi$ customers with unit revenue at most $f_u$, implying $V^{opt} \leq \Pi + f_u(L_1 + L_2 - \phi)$, and thus $\frac{V^{on}}{V^{opt}} \geq \frac{\alpha f_u(L_1 + L_2) + \Pi}{\Pi + f_u(L_1 + L_2 - \phi)} \geq \frac{\alpha f_u(L_1 + L_2)}{f_u(L_1 + L_2)} = \alpha$.

**Case 3**. $k < u < m$. In this case, $\Pi$ and $\phi$ are defined as the same with Case 2. Similarly, since *ONPS* reserves a capacity of $L_1 + L_2 - \Theta_k$ for all customers of classes $C_{k+1}$, ..., $C_m$, and a capacity of $L_2 - \Theta_u$ for all customers of classes $C_{u+1}$, ..., $C_m$, where $k + 1 \leq u \leq m - 1$, it is obvious that $V^{on} \geq \sum_{i=1}^{k} f_i(\Theta_i - \Theta_{i-1}) + f_{k+1}(\Theta_{k+1} - (\Theta_k - L_1)) + V + \Pi$ where $V = 0$ for $u = k + 1$ and $V = \sum_{i=k+2}^{u} f_i(\Theta_i - \Theta_{i-1})$ for $u > k + 1$. By Lemma 1, $V^{on} \geq \alpha(f_k L_1 + f_u L_2) + \Pi$. For *OPT*, it at best accepts all the $\phi$ customers with reservation price larger than $f_u$ and accepts the rest $L_2 - \phi$ customers with reservation price at most $f_u$ for resource $R_2$. Each customer assigned to resource $R_1$ is of unit revenue at most $f_k$. Therefore, $V^{opt} \leq \Pi + f_u(L_2 - \phi) + f_k L_1$, and thus $\frac{V^{on}}{V^{opt}} \geq \frac{\alpha(f_k L_1 + f_u L_2) + \Pi}{\Pi + f_u(L_2 - \phi) + f_k L_1} \geq \frac{\alpha(f_k L_1 + f_u L_2)}{f_u L_2 + f_k L_1} = \alpha$.

**Case 4**. $u = m$. Similar to the analysis in Case 3, by Lemma 1,

$V^{on} \geq \sum_{i=1}^{m} f_i(\Theta_i - \Theta_{i-1}) + f_{k+1}L_1 = \alpha(f_k L_1 + f_m L_2)$. For *OPT*, it at best gains a maximum reve-

nue of $V^{opt} = f_k L_1 + f_m L_2$, and thus $\frac{V^{on}}{V^{opt}} \geq \frac{\alpha(f_k L_1 + f_m L_2)}{f_k L_1 + f_m L_2} = \alpha$.

Combining the four cases, the revenue achieved by strategy *ONPS* is always no less than $\alpha$ times of that achieved by an optimal strategy, which gives us the theorem.

## An upper bound

In this section, we present an upper bound of competitive ratio for the online RM problem with two levels of resources and multiple fares as stated in Theorem 2, which implies that the online strategy *ONPS* proposed in the previous section is an optimal one.

**Theorem 2** *For the online quantity-based RM problem with two levels of resources and multiple fares, any online strategy has a competitive ratio of at most $\alpha$.*

**Proof of Theorem 2**. In order to derive a bound on the best-possible performance for an online strategy for the problem, we use the following extreme instances $I_i$ ($i = 1, \ldots, m$): in instance $I_i$ a sequence of $i(L_1 + L_2)$ customers, namely, $L_1 + L_2$ customers of each class $C_1$, $C_2$, $\ldots$, $C_i$ arrive in this order. The optimum revenues earned by an offline strategy *OPT* are $V^{OPT}(I_i) = f_i(L_1 + L_2)$ for $1 \leq i \leq k$ and $V^{OPT}(I_i) = f_k L_1 + f_i L_2$ for $k + 1 \leq i \leq m$, respectively.

For any online strategy *ON*, let $x_i^{ON}$ be the number of customers of class $C_i$ accepted by *ON* when presented with instance $I_m$. Note that *ON* has no way of knowing whether it is facing instance $I_i$ or some $I_j$ with $j > i$ before it has seen the first $i(L_1 + L_2)$ customers in the stated sequence. Therefore, for all $1 \leq j \leq i$, *ON* will accept exactly $x_j^{ON}$ customers of class $C_j$ when presented with instance $I_i$. Let $\omega_i = \Theta_i - \Theta_{i-1}$ for $i \neq k + 1$ and $\omega_{k+1} = \Theta_{k+1} + L_1 - \Theta_k$ where $\Theta_i$ is defined as the proposed online strategy *ONPS*. If $x_i^{ON} = \omega_i$ for all $i = 1, 2, \ldots, m$, then strategy *ON* is equivalent to *ONPS*, which implies that the competitive ratio of *ON* is just $\alpha$ by Theorem 1. Otherwise, if there exists some $x_i^{ON} \neq \omega_i$, letting $v = \min\{i | x_i^{ON} \neq \omega_i\}$, then there are two cases as discussed below according to the status of $v$.

**Case 1**. $x_v^{ON} < \omega_v$. In this case, to generate the worst algorithmic performance, the extreme instance $I_v$ will be presented for *ON*. Obviously, the revenue gained by *ON* facing $I_v$ is

$V^{ON} = \sum_{i=1}^{v} f_i x_i^{ON} < \sum_{i=1}^{v} f_i \omega_i$. Therefore, the best-possible performance of strategy *ON* is

$\frac{V^{ON}(I_v)}{V^{OPT}(I_v)} < \frac{\sum_{i=1}^{v} f_i \omega_i}{V^{OPT}(I_v)}$. Note that $V^{OPT}(I_v) = f_v(L_1 + L_2)$ for $1 \leq v \leq k$ and $V^{OPT}(I_v) = f_k L_1 + f_v L_2$ for

$k + 1 \leq v \leq m$. By Lemma 1 and the definition of $\omega_i$, $\frac{\sum_{i=1}^{v} f_i \omega_i}{V^{OPT}(I_v)} = \alpha$ for any $v$, and thus $\frac{V^{ON}(I_v)}{V^{OPT}(I_v)} < \alpha$.

**Case 2**. $x_v^{ON} > \omega_v$. Since $\omega_m \leq L_2 = \Theta_m$, it is obvious that $v < m$ in this case. Thus, there must be some $h$ ($0 \leq h < m$) satisfying $x_i^{ON} \geq \omega_i$ for $i = v, v + 1, \ldots, v + h$ while $x_{v+h+1}^{ON} < \omega_{v+h+1}$. The rest reasoning is similar to Case 1. Presenting the extreme instance $I_{v+h+1}$ for *ON*, it is also that $\frac{V^{ON}(I_{v+h+1})}{V^{OPT}(I_{v+h+1})} < \alpha$ in this case.

Combining Case 1 and Case 2, it is always that $V^{ON} \leq \alpha V^{OPT}$, which gives us Theorem 2.

## Application discussion

In this section, we present a numerical example in the field of airlines to illustrate the results of our model and to discuss the application of the proposed theoretical model. In fact, our method and model can also be applied in other fields, such as hotel booking with heterogenous rooms and car rental with different classes of autos.

Consider an airline that provides potential passengers with 80 economy class seats and 20 business class seats, for one flight. The full fares of economy tickets and business tickets are predetermined equal to $1000 and $2000, respectively. Additionally, there are two discount fares of economy tickets respectively equal to $500 and $750, and there is only one discount fare equal to $1600 for business seats. Following the theoretical model setting, it is that $L_1 = 80$, $L_2 = 20$, and thus $t = L_1/L_2 = 4$. Meanwhile, $f_1 = \$500$, $f_2 = \$750$, $f_3 = \$1000$, $f_4 = \$1600$, and $f_5 = \$2000$. According to the definition of the proposed online strategy, we further have $r_0 = 0$, $r_1 = 2/3$, $r_2 = 3/4$, $r_3 = 5/8$, $r_4 = 4/5$, $r_5 = 1$, and thus $\alpha = 0.59$, which means that any online seat allocation strategy cannot achieve a competitive ratio larger than 0.59 for this example.

In order to illustrate the results of our proposed model and online strategy, consider an extreme LBH customer arrivals stated as follows. In this LBH arriving process, a sequence of 100 customers of each class $C_1$, $C_2$, $C_3$, $C_4$, $C_5$ arrive in this order. Obviously, the optimal offline strategy with full information would only accept 100 customers of class $C_5$ and assign 20 of them to business seats charging $f_5$. The rest 80 customers are assigned to economy seats at $f_3$. The resulting maximum revenue is equal to $V^{opt} = 20f_5 + 80f_3 = \$120000$. For the same customer arriving sequence, the *first-come-first-served* strategy would accept 100 customers of class $C_1$ gaining $50000.

Following the online strategy *ONPS* proposed in our model, obviously $\Theta_1 = 59$, $\Theta_2 = 79$, $\Theta_3 = 93$, $\Theta_4 = 18$, and $\Theta_5 = 20$. That is, for the highest business class $C_5$, the proposed strategy *ONPS* defines a protection reservation capacity of 2 business seats. For the highest two business classes of customers, i.e. $C_4$ and $C_5$, 7 business seats are reserved as the protection capacity. For the highest economy class $C_3$, strategy *ONPS* defines a protection reservation capacity equal to 93-79 = 14. For the highest two economy class $C_2$ and $C_3$, strategy *ONPS* defines a protection reservation capacity equal to 93-59 = 34. For the lowest class $C_1$, *ONPS* sets a maximum number of accepted customers equal to 59. Therefore, facing the same instance, the proposed strategy *ONPS* will accept 59 customers of class $C_1$ and 20 customers of class $C_2$, and assign them to economy seats. 14 customers of class $C_3$ will be accepted and 13 of them will be assigned to business seats. The strategy *ONPS* will also accept 5 customers of class $C_4$ and 2 customers of class $C_5$, and assign them to business seats. The resulting revenue is equal to $59f_1 + 20f_2 + 14f_3 + 4f_4 + 2f_5 = \$70500$, which is more than that gained by the first-come-first-served strategy.

Obviously, the revenue gained by *ONPS* is 0.59 (= 70500/120000) times of that gained by an optimal offline strategy. Note that, the competitive ratio of the proposed strategy is just equal to $\alpha = 0.59$, which means that the presented extreme LBH instance is one of the worst cases for strategy *ONPS* from the perspective of competitive analysis of online algorithms. In other words, the practical performance will be better when facing other customer arriving instances.

Notice that, there is an implicit assumption on customer's reservation price in our model. We assume that the decision maker knows the customer's reservation price when she/he arrives. For some firm, in fact, it might analyze the psychological expectation of customer and draw the reservation price using big data technology. Even in the case where the firm cannot know customer's reservation prices, our model can also be used to guide the decisions on how many resources to be sold at each fare, especially to set limits for lower fare classes and to reserve capacity for higher fare classes, like the applications of theoretical models implied in [18].

## Conclusion

This paper studies an online revenue maximization problem of assigning heterogeneous customers to two levels of resources with capacity constraints. From the perspective of

competitive analysis, an upper bound for this online revenue management problem is derived and an online strategy whose competitive ratio matches the upper bound is also proposed. The model proposed in this paper generalizes the 2-level-2-class problem studied in [20] by considering multiple fare classes for each level of resources, i.e., the strategy as well as the theoretical result will degenerate to that of [20] if $k = 1$ and $m = 2$ in the current model. Besides, it will degenerate to the single-leg-multiple-fare problem studied in [18] when there is only one level of resources available.

There are several directions for future research. First, it is interesting to study the general problem with more than two levels of resources. Second, it is valuable to investigate the scenario with limited downgrading for customers with high reservation price. Third, it is also interesting and challenging to consider resources allocation problem together with pricing decisions.

## Acknowledgments

The author would like to thank the editor and four anonymous referees for their helpful comments and suggestions.

## Author Contributions

**Conceptualization:** Guanqun Ni.

**Formal analysis:** Guanqun Ni.

**Methodology:** Guanqun Ni.

**Writing – original draft:** Guanqun Ni.

**Writing – review & editing:** Guanqun Ni.

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
