## [Decision Letter · Decision Letter 0]

23 Aug 2022

PONE-D-22-19479A note on competitive analysis of online revenue management with two hierarchical resourcesPLOS ONE

Dear Dr. Ni,

Thank you for submitting your manuscript to PLOS ONE. After careful consideration, we feel that it has merit but does not fully meet PLOS ONE’s publication criteria as it currently stands. Therefore, we invite you to submit a revised version of the manuscript that addresses the points raised during the review process.

We look forward to receiving your revised manuscript.

Kind regards,

Ronnason Chinram, Ph.D.

Academic Editor

PLOS ONE

Journal Requirements:

2. Thank you for submitting the above manuscript to PLOS ONE. During our internal evaluation of the manuscript, we found significant text overlap between your submission and the following previously published works, some of which you are an author.

-https://www.sciencedirect.com/science/article/abs/pii/S0020019018302023?via%3Dihub

Please revise the manuscript to rephrase the duplicated text, cite your sources, and provide details as to how the current manuscript advances on previous work. Please note that further consideration is dependent on the submission of a manuscript that addresses these concerns about the overlap in text with published work.

"This work was partially supported by the MOE Layout Foundation of Humanities and Social Sciences under grant 20YJA630049 and Startup Program of Doctor Scientific Research in Shandong University of Technology."

"NO-The funders had no role in study design, data collection and analysis, decision to publish, or preparation of the manuscript."

Additional Editor Comments:

We inform you that your paper has received the major revision for the possible publication. Please see the comments of reviewers and kindly revise your paper and submit with the response letter at the earliest convenience.

Reviewers' comments:

Reviewer's Responses to Questions

**Comments to the Author**

1. Is the manuscript technically sound, and do the data support the conclusions?

Reviewer #1: No

Reviewer #2: Yes

Reviewer #3: Yes

Reviewer #4: Yes

2. Has the statistical analysis been performed appropriately and rigorously? 

Reviewer #1: No

Reviewer #2: Yes

Reviewer #3: Yes

Reviewer #4: N/A

3. Have the authors made all data underlying the findings in their manuscript fully available?

Reviewer #1: No

Reviewer #2: Yes

Reviewer #3: Yes

Reviewer #4: Yes

4. Is the manuscript presented in an intelligible fashion and written in standard English?

Reviewer #1: Yes

Reviewer #2: Yes

Reviewer #3: Yes

Reviewer #4: Yes

5. Review Comments to the Author

Reviewer #1: The paper studies a general and standard revenue management problems with different types of customers (who inheriate heterogeneity in their valuation on the services). A model is proposed to describe the pricing and system performance.

The problem is indeed important, while the paper fails to convince me its academic innovation and methodological contributions in the area of revenue management and pricing. In fact, there are a super rich of literature dealing with this problem and huge set of theoretical results are already presented. The modelling setting and assumptions adopoted in this paper seem to not the commonly used ones in literature in a negative way (for example, the paper assumes a segemented deterministic valuation on the service, not a distribution), which makes the results much less valueble and interesting.

In general, I suggest the authors to dive more into the rich literature and justify its academic innovation more clearly.

Reviewer #2: This paper studies an online revenue maximization problem of assigning heterogeneous customers to two levels of resources with capacity constraints. It extends the previous work in the literature with two classes of customers to a more general case with m (>1) classes of customers. From the perspective of competitive analysis, both an upper bound and an optimal online algorithm named ONPS (i.e., online nested protection strategy) are proposed. The topic is interesting, and the theoretical results are meaningful. The paper is also well organized in general.

There are some minor concerns as follows.

1) At the end of the 2nd last paragraph of the Introduction section, it is suggested to briefly explain and highlight the marginal contribution from the theoretical or practical perspective in this work.

2) In the 2nd paragraph of page 2, it seems that f_k is used to denote the price of the kth class of resources, while in Section 2 on page 4, f_i represents the reservation price of customers in class C_i. This may cause confusion. It is suggested to replace f_k with p_k or another notation in the former one.

3) The presentation of this work needs some improvement. For example,

(1) lines 109- 111 on page 5, “is measured by the ratio between the objective revenue achieved by the online strategy and the revenue by an optimal offline strategy, which is called…”.

(2) Line -7 on page 6, “The reasoning of part (2), However, is non-trivial. Firstly, …”

(3) It is suggested to delete words “we have” or “we have that” in many places in the context, such as in line -3 on page 6 and line 1 on page 7.

(4) Line 213 on page 10, which implies

(5) Line 252 on page 11, constraints

Reviewer #3: The authors provide an extension on an already published paper that deals with the competitive analysis of online revenue management with two hierarchical resources. Specifically, the authors generalize this concept to more than two classes of customers. The authors have provided the detailed theoretical analysis in order to support the extension. The provided mathematical analysis in the paper is detailed and correct and the authors have provided all the intermediate steps in order to enable the average leader to easily follow it. The authors should consider the following suggestions provided by the reviewer in order to improve the scientific depth of their manuscript, as well as they need to address the following comments in order to improve the quality of presentation of their manuscript. Initially, the title of the paper needs to be changed as it seems like the authors identify a mistake in the existing article based on the title that they have used. It needs to be clarified that this is an extension to an existing published paper where other authors have originally thought of novel ideas and problem formulations and this paper extends an already published research work. In Section 1, the authors need to better identify the research gap that exists in the literature as it seems that there are several practical based approaches, such as Grieco, Luigi Alfredo, et al., eds. Ad-Hoc, Mobile, and Wireless Networks: 19th International Conference on Ad-Hoc Networks and Wireless, ADHOC-NOW 2020, Bari, Italy, October 19–21, 2020, Proceedings. Vol. 12338. Springer Nature, 2020, that they deal with the same problem from a game theoretic point of view. The authors need to substantially revise the provided related work. In Section 2, the authors need to include at the very beginning a table summarizing the main notation that has been used in the paper which is extremely excessive in its current form. At the end of Section 3, the authors need to include an additional subsection discussing potential application fields where the proposed theoretical model can provide realistic solutions. Finally, the manuscript needs to be checked for typos, syntax, and grammar errors in order to improve the quality of its presentation.

Reviewer #4: Thank you for your interesting work and this note surely pushes the boundary of related research. A few comments:

1) On Page 2, second paragraph. It might be easier for readers if "sub-classes" are defined beforehand.

2) The connection to Ball and Queyranne was clearly stated; however, the explaination of the difference between this paper and B&Q, especially the fact that your two resources are "downgrade" compatible, unlike B&Q, can be clearer.

3) A possible improvement is to justify the fact that we know the customer's reservation price when he/she arrives.

A nicely written paper overall. Thank you for making the paper clear and readable. I enjoyed it a lot.

6. PLOS authors have the option to publish the peer review history of their article (what does this mean?). If published, this will include your full peer review and any attached files.

Reviewer #1: No

Reviewer #2: No

Reviewer #3: No

Reviewer #4: No

---

## [Author Response · Author response to Decision Letter 0]

5 Sep 2022

Dear Editor and Reviewers,

We greatly appreciate your precious comments and suggestions for our paper. We have carefully revised the manuscript as suggested. All changes have been highlighted in RED in our revision. Below are responses to the reviewers’ comments one-by-one in details.

Response to Reviewer #1:

Comment 1: The problem is indeed important, while the paper fails to convince me its academic innovation and methodological contributions in the area of revenue management and pricing. In fact, there are a super rich of literature dealing with this problem and huge set of theoretical results are already presented. In general, I suggest the authors to dive more into the rich literature and justify its academic innovation more clearly.

Response: Thank you very much for your comments. We have supplemented the literature (adding 10 references) and pointed out that the quantity-based resource allocation problem studied in our model is one of the key issues in the field of RM. However, the traditional models rely heavily on some restrictive and unrealistic assumptions about demand or arrivals. Besides, the previous studies usually assume that the resource is non-hierarchical, which is sometimes out of line with reality. Our main contribution is to relax these assumptions. We have summarized our contributions in the fourth paragraph of Introduction (lines 51-61 on page 2). We have also elaborated the differences and relationships between our paper and the previous studies, especially the most related works of Ball and Queyranne, Lan et al., and Ni et al. Please see the text highlighted in red in Introduction for details.

Comment 2: The modelling setting and assumptions adopted in this paper seem to not the commonly used ones in literature in a negative way (for example, the paper assumes a segmented deterministic valuation on the service, not a distribution), which makes the results much less valuable and interesting.

Response: Thanks a lot. First, we have taken seat allocation problem commonly faced by an airline as an example to illustrate that the proposed model is in line with reality. Please see the third paragraph of Introduction (lines 23-43 on page 2) for the example. Second, we have reformulated the concerned problem in the first paragraph of Problem Definitions (lines 113-124 on page 3), which defines both customer reservation price and customer classification more clearly and realistically.

Response to Reviewer #2:

General Comments: This paper studies an online revenue maximization problem of assigning heterogeneous customers to two levels of resources with capacity constraints. It extends the previous work in the literature with two classes of customers to a more general case with m (>1) classes of customers. From the perspective of competitive analysis, both an upper bound and an optimal online algorithm named ONPS (i.e., online nested protection strategy) are proposed. The topic is interesting, and the theoretical results are meaningful. The paper is also well organized in general. There are some minor concerns as follows.

Response: Thank you very much for your positive comments. We have carefully revised the paper according to your suggestions.

Comment 1: At the end of the 2nd last paragraph of the Introduction section, it is suggested to briefly explain and highlight the marginal contribution from the theoretical or practical perspective in this work.

Response: Thanks a lot. We have summarized our contributions in the fourth paragraph of Introduction (lines 51-61 on page 2) as you suggested.

Comment 2: In the 2nd paragraph of page 2, it seems that f_k is used to denote the price of the kth class of resources, while in Section 2 on page 4, f_i represents the reservation price of customers in class C_i. This may cause confusion. It is suggested to replace f_k with p_k or another notation in the former one.

Response: Thank you so much for this point. We have reformulated the concerned problem in the first paragraph of Problem Definitions (lines 113-124 on page 3), which defines fare class and customer reservation price more clearly, making the context consistent.

Comment 3: The presentation of this work needs some improvement. For example,

(1) Lines 109- 111 on page 5, “is measured by the ratio between the objective revenue achieved by the online strategy and the revenue by an optimal offline strategy, which is called…”

(2) Line 7 on page 6, “The reasoning of part (2), however, is non-trivial. Firstly …”

(3) It is suggested to delete words “we have” or “we have that” in many places in the context, such as in line 3 on page 6 and line 1 on page 7.

(4) Line 213 on page 10, which implies

(5) Line 252 on page 11, constraints

Response: Thank you very much for your detailed suggestions. All these comments have been modified and highlighted in red. Please see lines 141-142 on page 4, the middle of page 5, line 236 on page 8, and line 322 on page 9 for details.

Response to Reviewer #3:

General Comments: The authors provide an extension on an already published paper that deals with the competitive analysis of online revenue management with two hierarchical resources. Specifically, the authors generalize this concept to more than two classes of customers. The authors have provided the detailed theoretical analysis in order to support the extension. The provided mathematical analysis in the paper is detailed and correct and the authors have provided all the intermediate steps in order to enable the average leader to easily follow it. The authors should consider the following suggestions provided by the reviewer in order to improve the scientific depth of their manuscript, as well as they need to address the following comments in order to improve the quality of presentation of their manuscript.

Response: Thank you very much for your comments. We have carefully revised the paper according to your suggestions.

Comment 1: Initially, the title of the paper needs to be changed as it seems like the authors identify a mistake in the existing article based on the title that they have used. It needs to be clarified that this is an extension to an existing published paper where other authors have originally thought of novel ideas and problem formulations and this paper extends an already published research work.

Response: Thanks a lot. We have modified the title as you suggested.

Comment 2: In Section 1, the authors need to better identify the research gap that exists in the literature as it seems that there are several practical based approaches, such as Grieco, Luigi Alfredo, et al., eds. Ad-Hoc, Mobile, and Wireless Networks: 19th International Conference on Ad-Hoc Networks and Wireless, ADHOC-NOW 2020, Bari, Italy, October 19–21, 2020, Proceedings. Vol. 12338. Springer Nature, 2020, that they deal with the same problem from a game theoretic point of view. The authors need to substantially revise the provided related work.

Response: Thank you so much. We have supplemented the literature (adding 10 references) and elaborated the differences and relationships between our paper and the previous studies, especially the most related works of Ball and Queyranne, Lan et al., and Ni et al. Please see the text highlighted in red in Introduction for details. We have also carefully read and studies the 23 articles included in the mentioned proceedings. However, all the articles are of a very different topic and mainly deal with internet and network problem using methods of graph theory rather than game theory. Limited to our knowledge, we did not recognize the inherent relationship between these researches and ours. Would you please give us some clear indication of the concerned articles, such as their titles? If you are not satisfied with our responses, please give us more detailed comments and suggestions. We will try our best to improve the paper. Thanks again.

Comment 3: In Section 2, the authors need to include at the very beginning a table summarizing the main notation that has been used in the paper which is extremely excessive in its current form.

Response: Thanks a lot. According to your suggestion, we have added the Table 1 in Problem Definitions (lines 127-128 on page 4) listing the main symbols used in our model.

Comment 4: At the end of Section 3, the authors need to include an additional subsection discussing potential application fields where the proposed theoretical model can provide realistic solutions.

Response: Thanks a lot. We have discussed the applications of our model in a new section of Application Discussion (lines 268-319 on pages 8-9) as you suggested.

Comment 5: Finally, the manuscript needs to be checked for typos, syntax, and grammar errors in order to improve the quality of its presentation.

Response: Thank you so much for your suggestions. We have checked and revised our manuscript thoroughly. All changes have been highlighted in red in our revision.

Response to Reviewer #4:

General Comments: Thank you for your interesting work and this note surely pushes the boundary of related research. A nicely written paper overall. Thank you for making the paper clear and readable. I enjoyed it a lot.

Response: Thank you so much for your very kind comments. We have carefully revised the paper according to your suggestions.

Comment 1: On Page 2, second paragraph. It might be easier for readers if “sub-classes” are defined beforehand.

Response: Thanks a lot. We have restated the mentioned example and replaced “sub-classes” with “discount fares”. More details please see the third paragraph of Introduction (lines 23-43 on page 2).

Comment 2: The connection to Ball and Queyranne was clearly stated; however, the explanation of the difference between this paper and B&Q, especially the fact that your two resources are “downgrade” compatible, unlike B&Q, can be clearer.

Response: Thanks a lot. We have clearly explained the difference between B&Q and our model in the second to the last paragraph of Introduction (lines 88-94 on page 3). We have also pointed out that the traditional models usually assume the resource is non-hierarchical; however, our model deals with downgrade compatible hierarchical resources. More details please see again the third paragraph of Introduction (lines 23-43 on page 2).

Comment 3: A possible improvement is to justify the fact that we know the customer's reservation price when he/she arrives.

Response: Thank you very much. As you suggested, we have explained that the decision maker can draw the customer’s reservation price using big data technology, and pointed out that our model can also be used to guide the decisions on resource allocation in the case where the customer’s reservation price is unknown in the last paragraph of Application Discussion (lines 312-319 on page 9).

---

## [Decision Letter · Decision Letter 1]

10 Oct 2022

Competitive analysis of online revenue management with two hierarchical resources and multiple fare classes

PONE-D-22-19479R1

Dear Dr. Ni,

We’re pleased to inform you that your manuscript has been judged scientifically suitable for publication and will be formally accepted for publication once it meets all outstanding technical requirements.

Kind regards,

Ronnason Chinram, Ph.D.

Academic Editor

PLOS ONE

Additional Editor Comments (optional):

The revised manuscript has been well improved in presentation and organization. This paper is in scope of the journal and scientifically valid which is worthy of publication. It is my pleasure to explain that your paper has been received the accepted for the possible publication.

Reviewers' comments:

Reviewer's Responses to Questions

**Comments to the Author**

1. If the authors have adequately addressed your comments raised in a previous round of review and you feel that this manuscript is now acceptable for publication, you may indicate that here to bypass the “Comments to the Author” section, enter your conflict of interest statement in the “Confidential to Editor” section, and submit your "Accept" recommendation.

Reviewer #2: All comments have been addressed

Reviewer #3: All comments have been addressed

2. Is the manuscript technically sound, and do the data support the conclusions?

Reviewer #2: Yes

Reviewer #3: Yes

3. Has the statistical analysis been performed appropriately and rigorously? 

Reviewer #2: Yes

Reviewer #3: Yes

4. Have the authors made all data underlying the findings in their manuscript fully available?

Reviewer #2: Yes

Reviewer #3: Yes

5. Is the manuscript presented in an intelligible fashion and written in standard English?

Reviewer #2: Yes

Reviewer #3: Yes

6. Review Comments to the Author

Reviewer #2: The revised manuscript has been well improve in presentation and organization. There are no further comments.

Reviewer #3: The authors have addressed in detail the reviewers' comments. This reviewer has no further concerns regarding this paper.

7. PLOS authors have the option to publish the peer review history of their article (what does this mean?). If published, this will include your full peer review and any attached files.

Reviewer #2: No

Reviewer #3: No

---

## [Editor Report · Acceptance letter]

14 Oct 2022

PONE-D-22-19479R1 

Competitive analysis of online revenue management with two hierarchical resources and multiple fare classes 

Dear Dr. Ni:

I'm pleased to inform you that your manuscript has been deemed suitable for publication in PLOS ONE. Congratulations! Your manuscript is now with our production department. 

Kind regards, 

on behalf of

Dr. Ronnason Chinram 

Academic Editor

PLOS ONE